# Quantifying and understanding the triboelectric series of inorganic non-metallic materials

Haiyang Zou[1,5], Litong Guo[1,2,5], Hao Xue[1,3,5], Ying Zhang[1], Xiaofang Shen[3], Xiaoting Liu[3], Peihong Wang[1], Xu He[1], Guozhang Dai[1], Peng Jiang[1], Haiwu Zheng[1], Binbin Zhang[1], Cheng Xu[1,2] & Zhong Lin Wang[1,4 ✉]

Contact-electrification is a universal effect for all existing materials, but it still lacks a quantitative materials database to systematically understand its scientific mechanisms. Using an established measurement method, this study quantifies the triboelectric charge densities of nearly 30 inorganic nonmetallic materials. From the matrix of their triboelectric charge densities and band structures, it is found that the triboelectric output is strongly related to the work functions of the materials. Our study verifies that contact-electrification is an electronic quantum transition effect under ambient conditions. The basic driving force for contact-electrification is that electrons seek to fill the lowest available states once two materials are forced to reach atomically close distance so that electron transitions are possible through strongly overlapping electron wave functions. We hope that the quantified series could serve as a textbook standard and a fundamental database for scientific research, practical manufacturing, and engineering.

[1] School of Materials Science and Engineering, Georgia Institute of Technology, Atlanta, GA 30332-0245, USA. [2] School of Materials Science and Engineering, China University of Mining and Technology, Xuzhou 221116, People's Republic of China. [3] College of Materials, Xiamen University, Xiamen 361005, People's Republic of China. [4] Beijing Institute of Nanoenergy and Nanosystems, Chinese Academy of Sciences, Beijing 100083, People's Republic of China. [5]These authors contributed equally: Haiyang Zou, Litong Guo, Hao Xue ✉email: zhong.wang@mse.gatech.edu

The contact-electrification (CE) effect is a universal phenomenon that occurs for all materials, which refers to two materials that are electrically charged after physical contact. However, CE is generally referred to as triboelectrification (TE) in conventional terms. In fact, TE is a convolution of CE and tribology, while CE is a physical effect that occurs only due to the contact of two materials without rubbing against each other, and tribology refers to mechanical rubbing between materials that always involves debris and friction[1].

The key parameters for CE, the surface charge density, the polarity, and the strength of the charges, are strongly dependent on the materials[2–5]. The triboelectric series describes materials' tendency to generate triboelectric charges. The currently existing forms of triboelectric series are mostly measured in a qualitative method in the order of the polarity of charge production. Recently, a standard method[6] has been established that allows this material "gene" of triboelectric charge density (TECD) to be quantitatively measured by contacting a tested material with a liquid metal using the output of a triboelectric nanogenerator (TENG) under fixed conditions. A table has been set for over 55 different types of organic polymer films. In comparison, inorganic materials have different atomic structures and band structures from polymers; therefore, it is necessary to quantify the triboelectric series for a wide range of common solid inorganic materials and study their triboelectric series in order to establish a fundamental understanding about their underlying mechanisms.

One of the oldest unresolved problems in physics is the mechanism of CE[7,8]. Many studies have been done on the analysis of the amount of the generated charges, including the correlation of charge amount with chemical nature[2], electrochemical reactions[9], work function[10], ion densities[11], thermionic emission[9], triboemission[12,13], charge affinity[14], surface conditions and circumstances[15], and flexoelectricity[16]. These studies focus on certain samples and quantitative data measured under various environmental conditions. The sample difference and the variance in the measurement conditions would cause large errors, and the mechanism studies based on a small dataset may not be reliable enough to derive a general understanding of the phenomenon. A systematic analysis based on a high-quality quantified database acquired in a universal standard method with a large volume of samples would provide more accurate data and facilitate a comprehensive understanding of the relationship between CE and the materials' intrinsic properties.

Here, we applied a standard method to quantify the triboelectric series for a wide range of inorganic non-metallic materials. Nearly 30 common inorganic materials have been measured, and the triboelectric series is listed by ranking the TECDs. By comparing the work functions of these materials, we find that the polarity of the triboelectric charges and the amount of charge transfer are closely related to their work functions. The triboelectric effect between inorganic materials and a metal is mainly caused by electronic quantum mechanical transitions between surface states, and the driving force of CE is electrons seeking to fill the lowest available states. The only required condition for CE is that the two materials are forced into the atomically close distance so that electronic transitions are possible between strongly overlapping wave functions.

## Results

### The principles of measurement and experimental setup.
Non-metallic inorganics are mostly synthesized at high temperature, they are hard materials with high surface roughness, and it is a challenge to make an accurate measurement of the TECD between solid–solid interfaces due to poor intimacy with inaccurate atomic-scale contact. To avoid this limitation, we measured the TECD of the tested materials with liquid metal (mercury) as the contacting counterpart as we used for organic polymer materials[6]. The basic principle for measuring the TECD relies on the mechanism of TENG, which is shown in Fig. 1a–d. Details about the measurement technique and the experimental design as well as the standard experimental conditions have been reported previously[6]. The measurement method relies on the principle of TENG in contact-separation mode (Fig. 1b)[3,17]. When the two materials are separated, the negative surface charges would induce positive charges at the copper electrode side (Fig. 1c). When the gap distance reaches an appropriate distance $d_1$, charges fully transfer to balance the potential difference (Fig. 1d). When the tested material is pushed back in contact with liquid mercury, the charges flow back (Fig. 1e). The TECD is derived from the amount of charge flow between the two electrodes.

The tested materials were purchased from vendors or synthesized through a pressing and sintering process in our lab (Supplementary Table 1). The tested materials were carefully cleaned with isopropyl alcohol by cleanroom wipers and dried by an air gun. Then, the specimens were deposited by a layer of Ti (15 nm) and a thick layer of Cu (above 300 nm) at the back as an electrode, and have a margin size of 2 mm to avoid a short circuit when the sample contacts with mercury.

### The measured TECD.
One group of typical signals measured for mica–mercury are shown in Fig. 2. The open-circuit voltage reached up to 145.4 V (Fig. 2a). A total of 69.6 nC electrons (Fig. 2b) flowed between the two electrodes. For each type of material, at least three samples were measured to minimize the measurement errors. The results were recorded after the measured value reached its saturation level. This will eliminate the initial surface charges on the samples. Figure 2c shows the output of three samples of mica measured at different times, and the measured values have good repeatability (Fig. 2d) and stability.

The TECD refers to the transferred triboelectric charges per unit area of the CE surface. Nearly 30 kinds of common inorganic non-metallic materials were measured, and their triboelectric series is presented in Fig. 3. The quantified triboelectric series shows the materials' capabilities to obtain or release electrons during the CE with the liquid metal. We have introduced a normalized TECD $\alpha$ in our previous study

$$\alpha = \frac{\sigma}{|\sigma_{\text{PTFE}}|}, \tag{1}$$

where $\sigma$ is the measured TECD of material. Here, we keep using the same standard for these inorganic materials for reference, so that the values are comparable. The average TECD values and the normalized TECDs $\alpha$ of the measured materials are both listed in Table 1. The more negative the $\alpha$ value is, the more negative charges it will get from mercury, and vice versa. If two materials have a large difference of $\alpha$ values, they will produce higher triboelectric charges when rubbed together (Supplementary Fig. 1). In contrast, the less difference of $\alpha$ values, the fewer charges exchange between them. The triboelectric series is validated by cross-checking (Supplementary Figs. 2 and 3).

### Mechanism of CE for inorganic non-metallic materials.
The standard measurement quantifies the TECD of various materials, the obtained values are only dependent on the materials. It remains to be systematically investigated, such as why different materials have a different amount of charges transferred; why some materials will become positively charged, but others were negatively charged after contact and separation with the same material; why the polarity of charge can be switched when they were contacted with different materials.

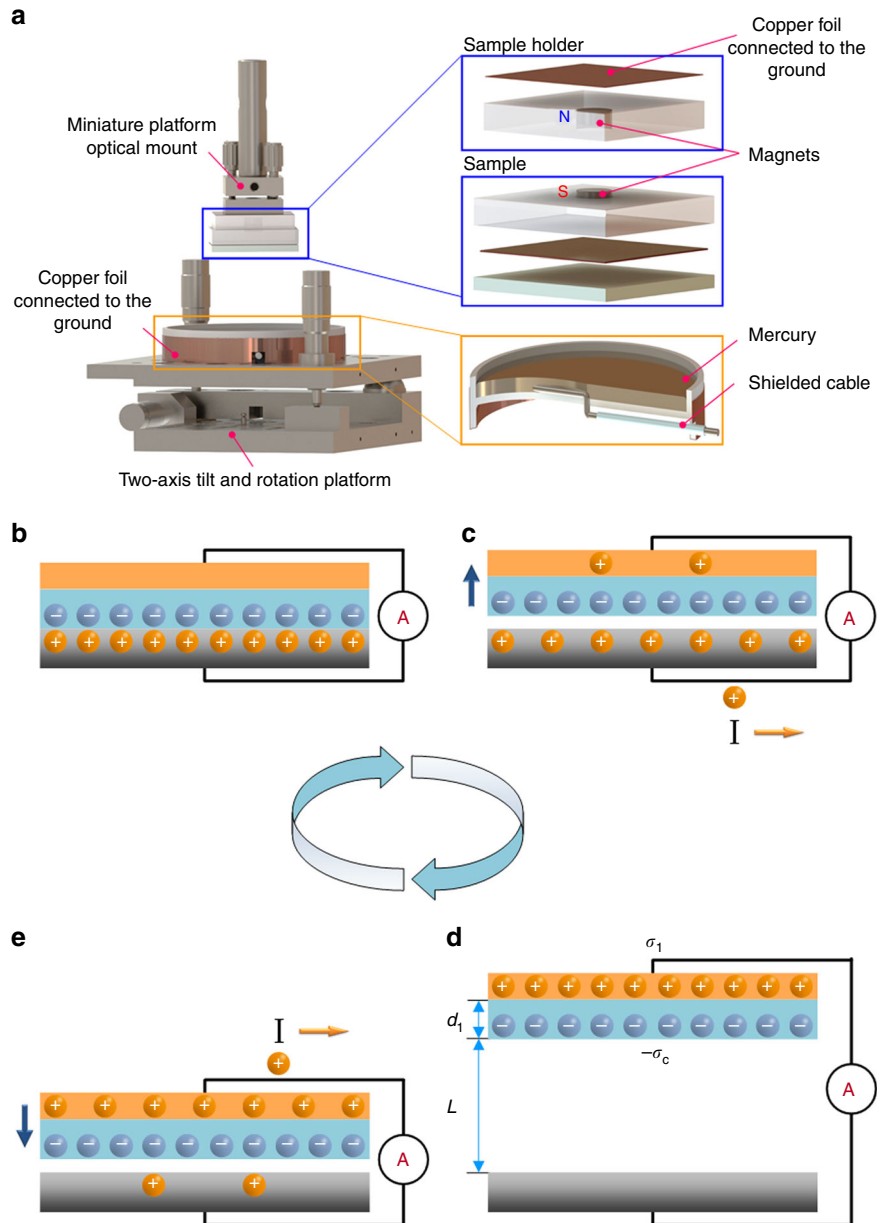

**Fig. 1 Experimental setup and the working mechanism of the measurement technique. a** Schematic diagram of the measurement system for the triboelectric charge density. **b–e** Schematic diagram of the mechanism for measuring the surface charge density. **b** Charges transferred between the two materials owing to the contact-electrification effect. There is no potential difference between the two materials when they are fully contacted with each other. **c** When the two materials are separated, the positive charges in mercury flow into the copper side in order to keep the electrostatic equilibrium. **d** When the gap goes beyond a specific distance L, there is no current flow between two electrodes. **e** When the material is in contact with mercury again, the positive charges flow from copper to mercury due to the induction of the negative charges on the surface of the inorganic material.

Here, we compare the TECD values with the relative work functions of the two contacting materials. In this study, all inorganic non-metallic materials were contacted with mercury. The work function of mercury is $\emptyset_{Hg} = 4.475\,\mathrm{eV}$[11]. The work functions of the tested materials are listed in Supplementary Table 2. The work functions of inorganic non-metallic materials are determined by materials themselves, but can be modified by crystallographic orientation, surface termination and reconstruction, and surface roughness, and so on. Therefore, some materials have a wide range of work functions in the literature. As shown in Fig. 4, as the work functions of materials decrease, the TECD values increase from $-62.66$ to $61.80\,\mu\mathrm{C\,cm^{-2}}$. The work function is related to the minimum thermodynamic energy needed to remove an electron from a solid to a point just outside the solid

surface. Our results show that electron transfer is the main origin of CE between solids and metal[18]. In addition, the polarity of the CE charges is determined by the relative work functions of materials. When the work function of the tested material **A** is smaller than the work function of mercury, $\emptyset_A < \emptyset_{Hg}$, the tested materials will be positively charged after intimate contact with mercury; when the work functions of tested material **B** are close to the work function of mercury, $\emptyset_B \approx \emptyset_{Hg}$, the tested material **B** will be little electrically charged; when the work functions of tested material **C** are larger than the work function of mercury, $\emptyset_C > \emptyset_{Hg}$, the tested materials will be negatively charged. The TECDs of tested materials are strongly dependent on the work function difference. If the two materials have a larger difference of work

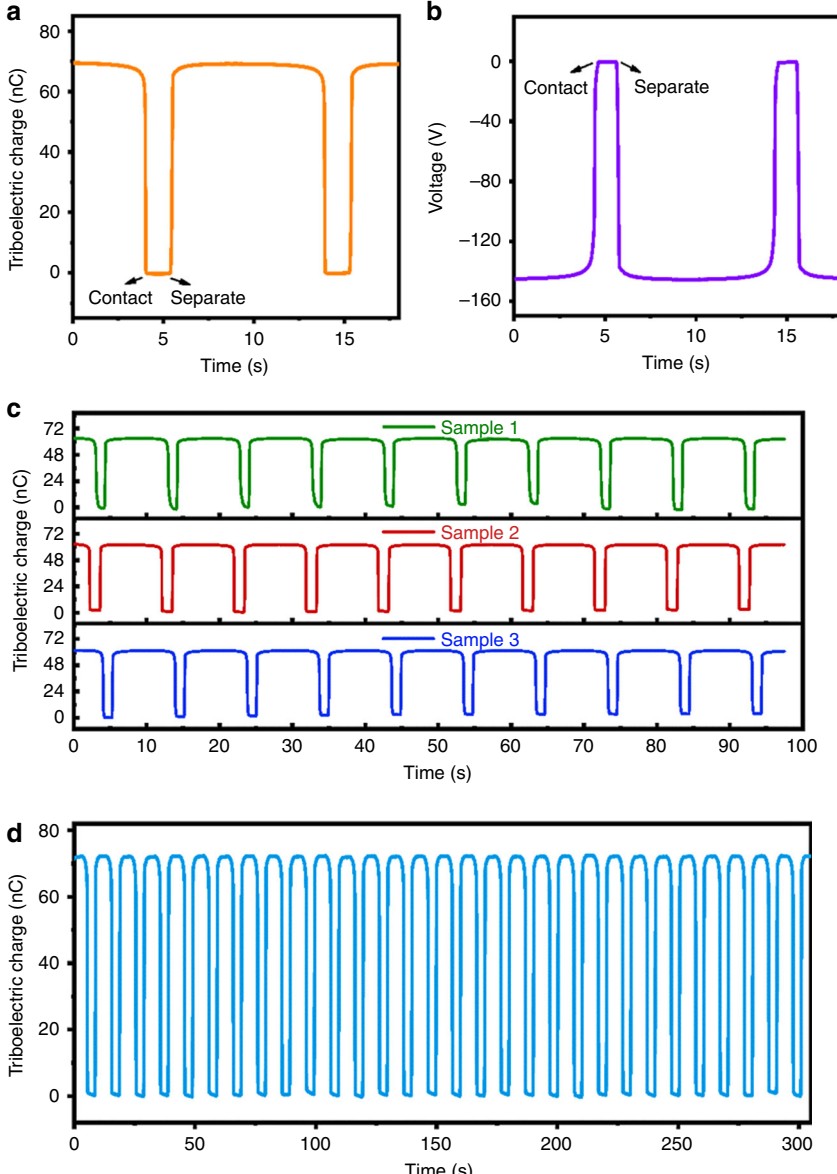

**Fig. 2 A set of typical measured signals of tested samples. a** Open-circuit voltage of mica during the processes of contact and separation with mercury.
**b** Curve of transferred charge between the two electrodes under short-circuit condition. **c** Measured charge transferred for three different samples of mica.
**d** Stability of the measured values for many cycles of operation. Source data are provided as a Source Data file.

functions, they will have more electrons transferred. These results show that electron transfer during CE is related to the band structure and energy level distribution. The electrons flow from the side that has high energy states to the side having low energy states.

The quantum mechanical transition model is proposed to explain the CE of inorganic non-metallic materials. Suppose we have a material **A**, which has a higher Fermi level than the Fermi level of the metal. The disruption of the periodic-potential function results in a distribution of allowed electric energy states within the bandgap, shown schematically in Fig. 5a, along with the discrete energy states in the bulk material. When the material is brought into intimate contact with the metal, the Fermi levels must be aligned (Fig. 5b), which causes the energy bands to bend and the surface states to shift as well. Normally, the energy states below the Fermi level of material **A**—$E_{FA}$ are filled with electrons and the energy states above $E_{FA}$ are mostly empty if the temperature is relatively low. Therefore, the electrons at the surface states above $E_{FA}$ will flow into the metal, thus the metal

gets negatively charged, and the originally neutralized material **A** becomes positively charged for losing electrons. The electrons that flowed from semiconductors or insulators to metals are mainly from the surface energy states. If the work functions of two materials (**B** and metal) are equal, there will be little electron transfer (Fig. 5c, d); therefore, it would have no electrification. When the work function of tested material **C** is lower than the work function of the metal (Fig. 5e), the Fermi levels tend to level, surface energy states shift down, and electrons flow reversely from metal to fill the empty surface states in material **C** to reach the aligned Fermi level (Fig. 5f). Thus, the tested material will be negatively charged and the metal becomes positively charged.

If two materials have a large difference of work functions, there are many discrete allowed surface states that electrons are able to transit; the surface is able to carry more charges after contact or friction. If the difference is low, few discrete surface states exist for electrons transition; the surface will be less charged. The surface charge density can be changed by contact with different

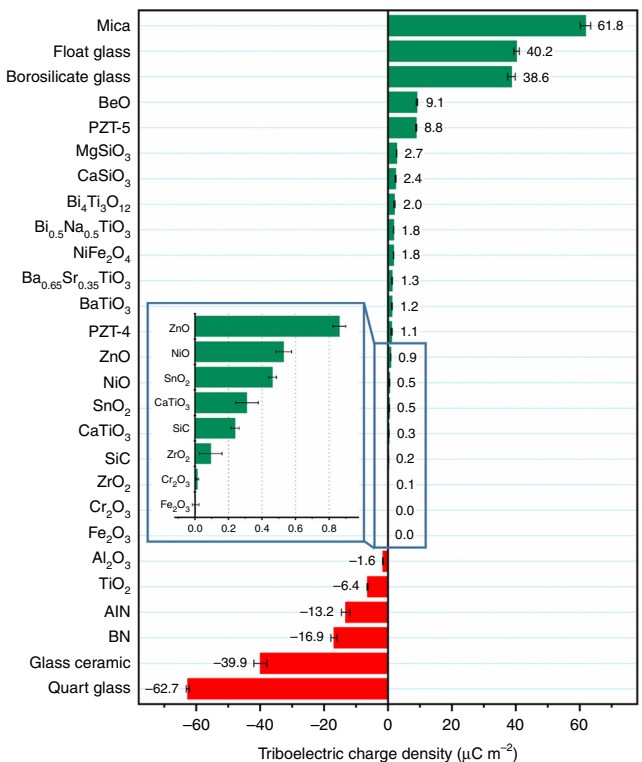

**Fig. 3 Quantified triboelectric series of some common inorganic non-metalic materials.** The error bar indicates the range within a standard deviation. Source data are provided as a Source Data file.

**Table 1 Triboelectric series of materials and their TECD.**

| Materials | Average TECD ($\mu C\ m^{-2}$) | STDEV | $\alpha$ |
|---|---|---|---|
| Mica | 61.80 | 1.63 | 0.547 |
| Float glass | 40.20 | 0.85 | 0.356 |
| Borosilicate glass | 38.63 | 1.18 | 0.342 |
| BeO | 9.06 | 0.21 | 0.080 |
| PZT-5 | 8.82 | 0.16 | 0.078 |
| $MgSiO_3$ | 2.72 | 0.07 | 0.024 |
| $CaSiO_3$ | 2.38 | 0.15 | 0.021 |
| $Bi_4Ti_3O_{12}$ | 2.02 | 0.21 | 0.018 |
| $Bi_{0.5}Na_{0.5}TiO_3$ | 1.76 | 0.05 | 0.016 |
| $NiFe_2O_4$ | 1.75 | 0.07 | 0.0155 |
| $Ba_{0.65}Sr_{0.35}TiO_3$ | 1.28 | 0.11 | 0.011 |
| $BaTiO_3$ | 1.27 | 0.08 | 0.0112 |
| PZT-4 | 1.24 | 0.12 | 0.011 |
| ZnO | 0.86 | 0.04 | 0.008 |
| NiO | 0.53 | 0.05 | 0.005 |
| $SnO_2$ | 0.46 | 0.02 | 0.004 |
| SiC | 0.31 | 0.07 | 0.003 |
| $CaTiO_3$ | 0.24 | 0.02 | 0.002 |
| $ZrO_2$ | 0.09 | 0.07 | 0.001 |
| $Cr_2O_3$ | 0.02 | 0.01 | 0.00013 |
| $Fe_2O_3$ | 0.00 | 0.02 | 0.000 |
| $Al_2O_3$ | −1.58 | 0.14 | −0.014 |
| $TiO_2$ | −6.41 | 0.18 | −0.057 |
| AlN | −13.24 | 1.35 | −0.117 |
| BN | −16.90 | 0.97 | −0.149 |
| Clear very high-temperature glass ceramic | −39.95 | 2.04 | −0.353 |
| Ultra-high-temperature quartz glass | −62.66 | 0.47 | −0.554 |

STDEV, standard deviation.
Note: The $\alpha$ refers to the measured triboelectric charge density of tested materials over the absolute value of the measured triboelectric charge density of the reference material (PTFE).

materials, due to the different levels of work functions. The polarity of surface charges can be switched as well, since they have different directions of electron transition.

For inorganic non-metallic materials, the dielectric constant is an important parameter. We have analyzed the relationship between dielectric constant and TECD. From the Gauss theorem, if we ignore the edge effect, the ideal induced short circuit transferred charge in the inorganic material–mercury TENG process is given by[6,17]:

$$Q_{SC} = \frac{S\sigma_c x(t)}{\frac{d_1 \varepsilon_0}{\varepsilon_1} + x(t)}, \qquad (2)$$

where $\varepsilon_1$ is the dielectric permittivity of the inorganic material, $d_1$ is the thickness, $x(t)$ is the separation distance over time $t$, and $\sigma_c$ is the surface charge density. From Eq. (1), under the measured conditions, $d_1 \ll x(t)$, and the part of $\frac{d_1 \varepsilon_0}{\varepsilon_1}$ can be ignored. Therefore, the dielectric constant will not influence the charge transfer $Q_{SC}$ and the surface charge density $\sigma_c$. As expected, the relation of TECD and dielectric constant of these materials is shown in Fig. 4; the measured TECDs are not affected by the dielectric constant of materials.

**Discussion**

A quantum mechanical transition always describes an electron jumping from one state to another on the nanoscale, while CE between solids is a macroscopic quantum transition phenomenon. Materials have a large scale of surface states to store or lose electrons, and charge transfer between two triboelectric materials is based on the capacitive model, so it can reach a significantly high voltage (>100 V)[19], which is different from the contact potential (mostly <1 V)[20]. The quantum transition model between the surface energy states explains how electrons are accumulated or released at the surfaces of inorganic dielectric materials and how

the surface becomes charged, while the contact potential model only explains carrier diffusion inside semiconductors[24]. The surface modification technologies, including impurity and doping elements, surface termination and reconstruction[21], surface roughness[22], and curvature effect[23] can tune the TECD. Based on the proposed model, it is suggested that the fundamental driving force of CE is that electrons fill the lowest available energy levels if there is little barrier. When the two materials have reached atomically close distance, electron transition is possible between strongly overlapping electron wave functions[25,26].

The work functions are determined by the compositions of compounds, chemical valence state, electronegativity[15], crystallographic orientation[27], temperature[19], defects[28,29], and so on. Accordingly, the calculation of work functions can be used as a comparison of a materials' property of TE and to estimate their triboelectric output. In addition, the work functions can be modified to improve the TE for enhancing the triboelectric effect for energy harvesting[30–33] and sensing[34,35], or reduce the electrical discharge due to CE to improve safety.

In summary, we have quantitatively measured the triboelectric series of some common inorganic non-metallic materials under defined conditions. The TECD data obtained depends only on the nature of the material. This serves as a basic data source for investigating the relevant mechanism of CE, and a textbook standard for many practical applications such as energy harvesting and self-powered sensing. The study verifies that the electron transfer is the origin of CE for solids, and that CE between solids is a macroscopic quantum mechanical transition effect that electrons transit between the surface states. The driving force for CE is that electrons tend to fill the lowest available surface states. Furthermore, the TE output could be roughly estimated and compared by the calculation of work functions, and ajusted by the modification of the material's work function through a variety of methods.

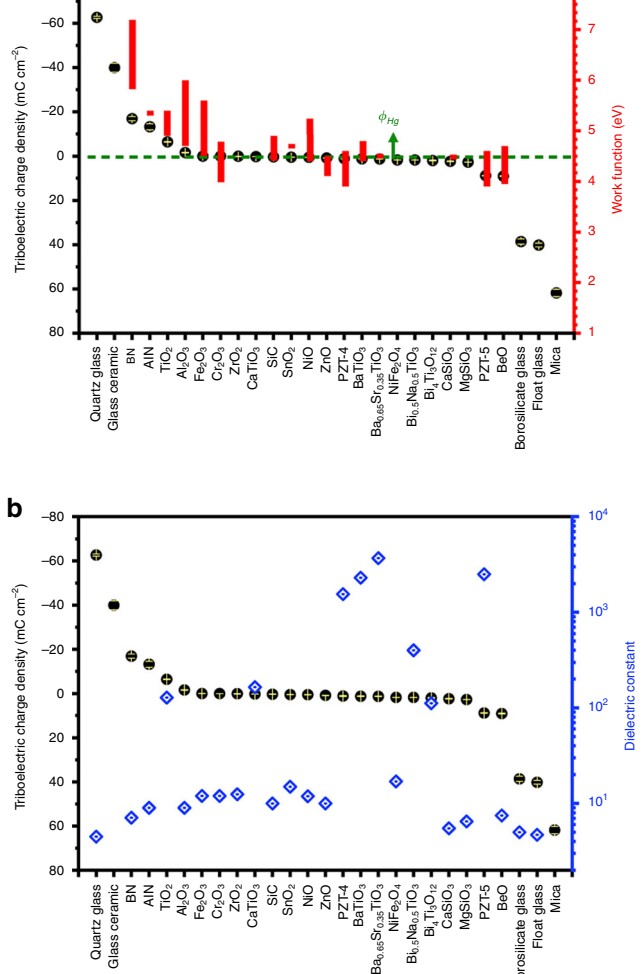

**Fig. 4 The influence of work function and dielectric constant on contact-electrification. a** Relationship between the triboelectric charge density and work functions of materials. **b** Relationship between the triboelectric charge density and dielectric constant. Source data are provided as a Source Data file.

## Methods

**Sample preparation**. The tested materials were purchased from vendors or synthesized through a pressing and sintering process. Some of the ceramic specimens, such as $MgSiO_3$, $CaSiO_3$, $Bi_4Ti_3O_{12}$, $Bi_{0.5}Na_{0.5}TiO_3$, $NiFe_2O_4$, $Ba_{0.65}Sr_{0.35}TiO_3$, $BaTiO_3$, and $CaTiO_3$, were prepared using a conventional solid-state reaction and solid-phase sintering. Some materials, such as ZnO, NiO, $SnO_2$, $Cr_2O_3$, $Fe_2O_3$, and $TiO_2$, were prepared by solid-phase sintering method using commercial ceramic powders. The details were described below.

For $MgSiO_3$, the high-purity MgO (99.5%) and $SiO_2$ (99.5%) powders were baked at 80 °C for 5 h to remove hygroscopic moisture and mixed in an ethanol medium by ball milling for 8 h according to the stoichiometric formula. The slurry was dried at 110 °C for 10 h and the dried powder was calcined at 1100 °C for 3 h, and then ball-milled in an ethanol medium for 8 h. After drying again, the obtained powders were granulated with polyvinyl alcohol as a binder and pressed into green disks with a diameter of 2 in. and a thickness of 1 mm under a pressure of 30 MPa. Next, the green disks were heated at 600 °C for 3 h to remove the binder, and then sintered at 1400 °C for 2 h. After the obtained ceramic disks were polished on both sides, the gold electrode was sputtered on one side.

Other samples, including $CaSiO_3$, $Bi_4Ti_3O_{12}$, $Bi_{0.5}Na_{0.5}TiO_3$, $NiFe_2O_4$, $Ba_{0.65}Sr_{0.35}TiO_3$, $BaTiO_3$, and $CaTiO_3$, are prepared similarly to $MgSiO_3$, except that there are differences in the temperature and holding time of powder calcination and ceramic sintering. Specific parameters for different samples are listed in the Supplementary Table 1.

For single element oxide, including ZnO, NiO, $SnO_2$, $Cr_2O_3$, $Fe_2O_3$, and $TiO_2$, the samples are directly prepared by solid-phase sintering method using commercial powders as the raw materials. Taking zinc oxide as an example, the high-purity ZnO powders (99.5%) were granulated with polyvinyl alcohol as a binder and pressed into green disks with a diameter of 2 in. and a thickness of 1 mm under a pressure of 30

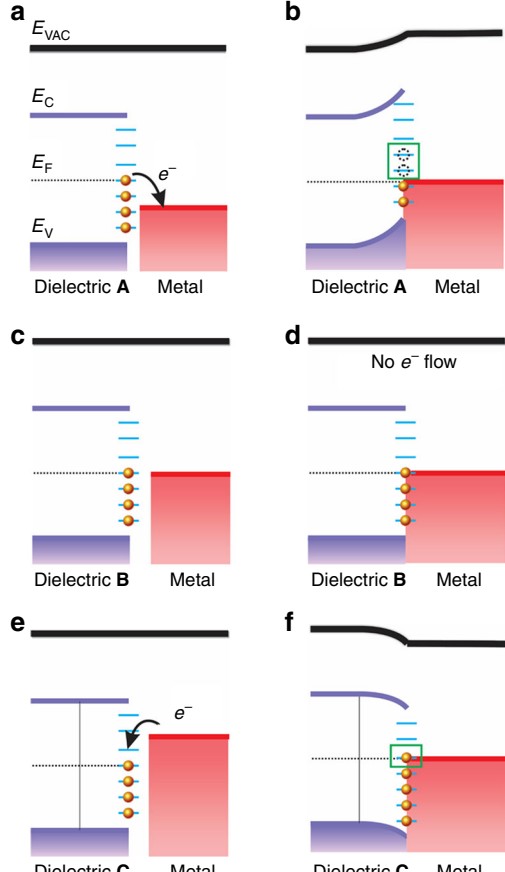

**Fig. 5 Electronic quantum transition model of contact-electrification between a dielectric and metal. a** When a dielectric **A** is brought into contact with the metal as shown in the figures, some electrons on the surface states flow into metal to seek the lowest energy states. **b** The energy bands bend to align the Fermi levels. Most electrons at the surface energy states above the balanced Fermi level flow into metal and left an equal amount of holes at the surface (as shown in green box). Thus, the original neutrally charged dielectric **A** turns to have positive charges on the surfaces due to the electrons lose. **c**, **d** When a dielectric **B** is brought into contact with the metal, the Fermi levels are balanced, the surface energy states equal. There are no quantum transitions between the two materials. **e** When a dielectric **C** contacts the metal, electrons on the surface of the metal flow into the dielectric **C** to seek the lowest energy levels. **f** The energy bands shift to align the Fermi levels. Electrons flow from metal to dielectric **C** to fill the empty surface states due to the difference of energy levels (as shown in the green box). The original neutrally charged dielectric **C** turns to carry negative charges on the surfaces by obtaining electrons.

MPa. Next, the green disks were heated at 600 °C for 3 h to remove the binder, and then sintered at 1200 °C for 1.5 h. After the obtained ceramic disks were polished on both sides, the gold electrode was sputtered on one side.

Samples, such as AlN, $Al_2O_3$, BeO, mica, float glass, borosilicate glass, PZT-5, SiC, $ZrO_2$, BN, clear very high-temperature glass ceramic, and ultra-high-temperature quartz glass, were directly purchased from different companies, which were also listed in the Supplementary Table 1.

The materials were washed with isopropyl alcohol, cleaned with cleanroom wipers, and dried by an air gun. Then, the materials were deposited with a layer of Ti (10 nm) and a thick layer of copper (above 300 nm) with a margin size of 2 mm by E-beam evaporator (Denton Explorer).

**The measurement of TECDs**. The samples were placed on the linear motor and moved up and down automatically with the help of the linear motor control program and system. For some inorganic compounds, the TECDs are relatively small; the turbulent caused by the motion of tested samples would cause some noise because of the friction between the platinum wire and mercury. Therefore,

the platinum wire was then designed to go through the bottom of the Petri dish and fully immersed in the liquid metal, and sealed by epoxy glue. In this way, there is no contact and separation between them; therefore, the noise is minimized.

The sample's surfaces were carefully adjusted to ensure the precisely right contact between the tested material and the liquid mercury. The position and angles were adjusted by a linear motor, a high load lab jack (Newport 281), and a two-axis tilt and rotation platform (Newport P100-P). The short-circuit charge $Q_{SC}$ and open-circuit voltage $V_{OC}$ of the samples were measured by a Keithley 6514 electrometer in a glove box with an ultra-pure nitrogen environment (Airgas, 99.999%). The environmental condition was fixed at 20 ± 1 °C, 1 atm with an additional pressure of 1–1.5 in. height of $H_2O$ and 0.43% relative humidity. In addition, samples were kept in the glove box overnight to eliminate the water vapor on the surface of the samples.

## Data availability

The datasets generated during and/or analyzed during the current study are available from the corresponding author. The source data underlying Figs. 2a–d, 3, and 4a–b are provided as a Source Data file.

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

## Acknowledgements

This work was financially supported by the Hightower Chair Foundation in Georgia Tech.

## Author contributions

Z.L.W. supervised and guided the project; H. Zou, L.G., H.X., and X.H. fabricated the devices; H.X., X.S., and X.L. synthesized the materials; H. Zou, Y.Z., and P.W. designed the measurement; L.G., H.X., H. Zou, G.D., P.J., B.Z., C.X., and H.Z. performed the experiments; L.G., H. Zou, and H.X. analyzed the data, H. Zou proposed the model; the manuscript was prepared with input from all authors.

## Competing interests

The authors declare no competing interests.
