## [Peer Review File · Nature Communications]

Reviewers' Comments:

Reviewer #1:

Remarks to the Author:

In this paper, the authors used a developed standard method to quantify the triboelectric series of inorganic nonmetallic materials. This work establishes a fundamental material's property of quantitative triboelectrification for the first time. In addition, from the measured data, they further investigated the basic principles of the contact-electrification (CE), and verified the origin of CE. These are great breakthroughs for understanding the triboelectric series of solid materials. This paper not only provides an important database for manufacturing and engineering, but also deeply studied the fundamental science and reveal the secret of an old unresolved problems in physics. Therefore, I strongly recommend the manuscript to be published in Nature Comm.

There are several questions to clear:

1. How this setup avoids the initial surface charge since the materials may contact other materials or air, and they will carry different surface charges before CE.
2. In the manuscript, the author seems not mentioned the size effect. Will different sizes affect the measuring results?
3. Can the authors use air instead of ultra-pure nitrogen to monitor the ambient environment?
4. The quantum transition model well explained the fundamental questions. It would be better for the authors to describe more details about the difference between the quantum transition model and contact potential theory. Why contact potential has a low voltage, but the contact electrification has such a large voltage? Although the authors have point out some, they may need more words to help the readers for better understanding.

By reading through the paper, I am pleased to learn the demonstrations and theory model presented. I believe this work can have a significant impact on this area, and the model will be widely used to study the triboelectrification between different materials.

Reviewer #2:

Remarks to the Author:

This work is an extended study of the author's previous work. In this work, they applied the previously developed method to understand CE of inorganic non-metallic materials. The results reported in this work can be very useful database for further advancement of triboelectric energy harvesting devices, sensors, or other applications utilizing triboelectric effect. Thus, I would like to suggest the publication of this work and further investigation on other triboelectric charging mechanisms in the future.

One concern I have in this work is the following. The proposed method agrees pretty well with most of inorganic materials listed in Fig. 3. However, the charge density difference for the materials ranging from Fe₂O₃ to BaTiO₃ (materials in the blue line box) is very small to make a general conclusion. For instance, the difference between the same materials, which may be originated from different surface states density of the source materials, might be bigger than that between different materials. Thus, I wonder if it is reasonable to list the materials in the box in order. I also wonder how many samples were used for this and if the work function of the listed materials are directly measured or are the values that are reported previously.

Reviewer #3:

Remarks to the Author:

Triboelectric nanogenerators (TENGs) are of wide interest, and one of the fundamental challenges is to

quantify the triboelectric characteristics of emerging materials, and rank them in the so-called triboelectric series. The authors previously published a liquid mercury based method to quantify triboelectric materials [Nat Commun. 2019; 10: 1427], in which the liquid metal serves as one of the TENG electrodes. While the current work is useful, it largely builds on "exactly the same method" published by the authors, and is severely lacking novelty. As such, this reviewer does not find the present work suitable for publishing in Nature Communications. The present manuscript is also riddled with typos, and some of the sentences are borrowed from their previous publication [Nat Commun. 2019; 10: 1427]. Few examples include the following:

- Line 26 Abstract: The dual use of "inorganic" in "...inorganic nonmetallic inorganic materials..." .
- Line 136: The units for the work function of mercury is missing.
- Line 162: The notation EFA is not defined.
- Figure 5b is not cited in the text.

Reviewer #1 (Remarks to the Author):

In this paper, the authors used a developed standard method to quantify the triboelectric series of inorganic nonmetallic materials. This work establishes a fundamental material's property of quantitative triboelectrification for the first time. In addition, from the measured data, they further investigated the basic principles of the contact-electrification (CE), and verified the origin of CE. These are great breakthroughs for understanding the triboelectric series of solid materials. This paper not only provides an important database for manufacturing and engineering, but also deeply studied the fundamental science and reveal the secret of an old unresolved problems in physics. Therefore, I strongly recommend the manuscript to be published in Nature Comm.

Response: We really appreciated the time and good comments from the reviewer. Thanks for the questions and suggestions.

There are several questions to clear:

1. How this setup avoids the initial surface charge since the materials may contact other materials or air, and they will carry different surface charges before CE.

Response: Thanks for the opinion from the reviewer. Through experiment results, we could find the initial charges cannot be kept all the time, will be lost during the operation, the final measured charge density will be saturated under the measured conditions, and the initial charges will not influence the measured charge density value. To eliminate the initial surface charges of the samples, each sample had been kept running until it reaches its saturation level, the results were recorded after the measured value reach its steady state (e.g. no obvious change after a hundred cycles).

The initial charges only influence the signals at the beginning, and the tested materials will have a stable output value of TECD after running for many cycles no matter what the initial charges are high or low. This may because the initial charges would be consumed by the impedance of the devices and measurement system after many cycles.

2. In the manuscript, the author seems not mentioned the size effect. Will different sizes affect the measuring results?

Response: We thank the question from the reviewer. Previously, we have found that the size would not have a large influence on the measured values. We fabricated devices with different sides of 1.5 in \times 1.5 in, and 1 in \times 1 in (both have a 2 mm edge to avoid the short circuit). The

measuring results are shown in Fig. R1. The calculated TECD for the two devices are about 7.02 nC cm^{-2} , and 6.80 nC cm^{-2} . The results are very close, which is within the measurement error.

Fig. R1 | The triboelectric charge transferred for samples with different size.

3. Can the authors use air instead of ultra-pure nitrogen to monitor the ambient environment?

Response: We thank the question from the reviewer. To measure the TECD of various materials, it should build up a standard environment, that all conditions should be stable with limit variances. If the measurement is under the air gas, the mercury would be oxide easily. Also, according to the Paschen's law, the highest spark over potential is obtained with nitrogen, which could prevent arc-over between the two materials. Therefore, we believe the usage of ultra-fine nitrogen can set up more stable conditions for the measurement.

4. The quantum transition model well explained the fundamental questions. It would be better for the authors to describe more details about the difference between the quantum transition model and contact potential theory. Why contact potential has a low voltage, but the contact electrification has such a large voltage? Although the authors have point out some, they may need more words to help the readers for better understanding.

Response: We thank the reviewer's suggestion. The contact potential is mostly $< 1\text{V}$, and the CE usually have a significant high value of voltage ($>100\text{ V}$). Obviously, the contact potential can not explain the voltage difference. It has been approved that the contact electrification is based on capacitive model, that charges are transferred through external circuit. The contact potential model describe the carrier diffusion in semiconductors, which only includes one physical process. The quantum transition model describe the electron transition and electron store, which well

explained why the surface of materials have surface charges after CE. The reason why CE has a large voltage may due to the large surface area and many surface states to store the electrons.

As the reviewer suggested, some details have been added in the manuscript.

Fig. R2 | Operation of the triboelectric Nanogenerator based on contact and separation model in open-circuit condition and short-circuit condition. ^[1]

[1] Z. L. Wang, L. Lin, J. Chen, S. Niu, Y. Zi, *Triboelectric Nanogenerators*, Springer International Publishing, Berlin 2016.

By reading through the paper, I am pleased to learn the demonstrations and theory model presented. I believe this work can have a significant impact on this area, and the model will be widely used to study the triboelectrification between different materials.

Response: We thanks for the reviewer's time and efforts on our work.

Reviewer #2 (Remarks to the Author):

This work is an extended study of the author's previous work. In this work, they applied the previously developed method to understand CE of inorganic non-metallic materials. The results reported in this work can be very useful database for further advancement of triboelectric energy harvesting devices, sensors, or other applications utilizing triboelectric effect. Thus, I would like to suggest the publication of this work and further investigation on other triboelectric charging mechanisms in the future.

Response: We thank the time and good comments from the reviewer.

One concern I have in this work is the following. The proposed method agrees pretty well with most of inorganic materials listed in Fig. 3. However, the charge density difference for the materials ranging from Fe₂O₃ to BaTiOs (materials in the blue line box) is very small to make a general conclusion. For instance, the difference between the same materials, which may be originated from different surface states density of the source materials, might be bigger than that between different materials. Thus, I wonder if it is reasonable to list the materials in the box in order. I also wonder how many samples were used for this and if the work function of the listed materials are directly measured or are the values that are reported previously.

Response: Thanks for the good question from the reviewer. We agree with the reviewer that there are many other factors which can alter the order of materials listed in the triboelectric series slightly, such as the surface states, doping, impurities, and structure etc. But the values and positions in this table are well enough to indicate the order of these materials under the certain test conditions to reflect the intrinsic property of such materials. Is Cr₂O₃ always have larger negative charge than Fe₂O₃? No, we do not think so either. For the materials from different vendors or synthesized in different methods, the TECD values will vary slightly in a small range. This range will cause the alternation of the order in the box, but this triboelectric series still give a reference number to quantitatively value the triboelectric property of tested materials. For example, we set up a table for listing the mechanical properties of different steels, but they still varies with different compositions or manufactured at different conditions, the table would provide a reference for the engineers in industry. We list them in the box so that the readers are able to easily tell the difference.

For each type of material, at least three samples were measured to minimize the measurement errors. The same type of materials are from the same vendors or synthesized in the same method. If the difference of the measured values is large, we will measure more samples to avoid the measurement error. Direct measurement of the work functions is indeed a good

suggestion. Here, we mainly use the data reported in literature previously. We think this is a more general feasible method.

Reviewer #3 (Remarks to the Author):

Triboelectric nanogenerators (TENGs) are of wide interest, and one of the fundamental challenges is to quantify the triboelectric characteristics of emerging materials, and rank them in the so-called triboelectric series. The authors previously published a liquid mercury based method to quantify triboelectric materials [Nat Commun. 2019; 10: 1427], in which the liquid metal serves as one of the TENG electrodes. While the current work is useful, it largely builds on "exactly the same method" published by the authors, and is severely lacking novelty. As such, this reviewer does not find the present work suitable for publishing in Nature Communications.

Response: We thanks for the reviewer's time and efforts on our work, and the comments from the reviewer. Previously, we have published a standard method for measuring the property of triboelectrification and the triboelectric series of **various polymers** has been quantified. As a sister chapter of the previous report, here, by using the established measurement method, for the first time, we quantified the triboelectric series for near **30 inorganic nonmetallic materials** in a universal standard method; by using the measured high-quality quantitative data, we systematically studied the property of materials' triboelectrification and their mechanisms. The data for inorganic nonmetallic materials are for the first time, and they will be included in future text books. The creative points are as follows.

1. It verifies that the origin of CE for solids is electron transfer;
2. It verifies the quantum mechanical transition model, and CE is a macroscopic quantum transition phenomenon;
3. It verifies that the driving force of CE for solids is electrons seek to fill the lowest available states;
4. The quantum mechanical transition model well explains the fundamental questions: why the triboelectrification is material dependent; why they will be positively or negatively charged; why the polarity and amount of charges changes after contacting with different materials; how electrons are accumulated and released at the surface of dielectrics; why doping, impurity, surface modification changes their triboelectrification;

Above all, our paper not only used the standard method to measure the inorganic nonmetallic materials, but also study the mechanism of contact electrification by the big data measured. Triboelectrification is a key material's genome, and every material exhibits triboelectrification. However, its mechanism is still one of the oldest unresolved problems in

physics^[1,2]. There are been many different theoretical models for the discussion of the principle of contact-electrification based on individuals or a small number of materials and their qualitative data under various conditions. The results are affected by many environmental factors, as well as individual differences, therefore the conclusion may not accurately reflect most materials and the fundamental mechanism.

We systematically studied the property of materials' triboelectrification and their mechanisms by using the measured high-quality quantitative data. The first quantified triboelectric series of inorganic nonmetallic materials would certainly provide a better indication of the charging behavior during the CE process. From the study, the triboelectrification behavior can be predicted and compared by the calculation of work function and the CE process can be tuned by modification of the work function in plenty of methods. This will be a textbook standard for scientific research and practical manufacturing and engineering for improving the output of triboelectric nanogenerator (TENG) or reducing the electrical discharge due to the CE for safety issues.

The present manuscript is also riddled with typos, and some of the sentences are borrowed from their previous publication [Nat Commun. 2019; 10: 1427]. Few examples include the following:

- **Line 26 Abstract: The dual use of "inorganic" in "...inorganic nonmetallic inorganic materials..."**
- **Line 136: The units for the work function of mercury is missing.**
- **Line 162: The notation EFA is not defined.**
- **Figure 5b is not cited in the text.**

Response: Thank you very much to the reviewer for their careful reading and suggestions. This is very helpful for us to modify the article and present the article in high quality. We have carefully revised the article. The mentioned typos have been modified, and the other contents have been carefully checked. We hope the revised manuscript could meet standard for acceptance.

[1] B. D. Terris, J. E. Stern, D. Rugar, H. J. Mamin, *Phys. Rev. Lett.* **1989**, 63, 2669.

[2] P. E. Shaw, *Nature* **1926**, 118, 659.

Reviewers' Comments:

Reviewer #1:

Remarks to the Author:

The authors did respond to all the doubts well. The manuscript is also presenting in good order. I recommend publishing this manuscript in the current form.

Reviewer #2:

Remarks to the Author:

The authors answered most of questions raised by the reviewer.

REVIEWERS' COMMENTS:

Reviewer #1 (Remarks to the Author):

The authors did respond to all the doubts well. The manuscript is also presenting in good order. I recommend publishing this manuscript in the current form.

Response: We appreciate the good comment from this reviewer.

Reviewer #2 (Remarks to the Author):

The authors answered most of questions raised by the reviewer.

Response: We thanks the time and effort from the reviewer.